# DNA Methylation Is a Potential Biomarker for Cardiometabolic Health in Mexican Children and Adolescents

Abeer A. Aljahdali [1,2], Jaclyn M. Goodrich [3,*], Dana C. Dolinoy [2,3], Hyungjin M. Kim [4], Edward A. Ruiz-Narváez [2], Ana Baylin [2,5], Alejandra Cantoral [6], Libni A. Torres-Olascoaga [7], Martha M. Téllez-Rojo [7] and Karen E. Peterson [2,3]

1.  Department of Clinical Nutrition, King Abdulaziz University, Jeddah 21589, Saudi Arabia
2.  Department of Nutritional Sciences, University of Michigan, Ann Arbor, MI 48109, USA
3.  Department of Environmental Health Sciences, University of Michigan, 1415 Washington Heights, Ann Arbor, MI 48109, USA
4.  Center for Computing, Analytics and Research, University of Michigan, Ann Arbor, MI 48109, USA
5.  Department of Epidemiology, University of Michigan, Ann Arbor, MI 48109, USA
6.  Department of Health, Iberoamericana University, Mexico City 01219, Mexico
7.  Center for Nutrition and Health Research, National Institute of Public Health, Cuernavaca 62100, Mexico
*   Correspondence: gaydojac@umich.edu; Tel.: +1-(734)-647-4564

**Abstract:** DNA methylation (DNAm) is a plausible mechanism underlying cardiometabolic abnormalities, but evidence is limited among youth. This analysis included 410 offspring of the Early Life Exposure in Mexico to Environmental Toxicants (ELEMENT) birth cohort followed up to two time points in late childhood/adolescence. At Time 1, DNAm was quantified in blood leukocytes at long interspersed nuclear elements (LINE-1), *H19*, and 11β-hydroxysteroid dehydrogenase type 2 (*11β-HSD-2*), and at Time 2 in peroxisome proliferator-activated receptor alpha (*PPAR-α*). At each time point, cardiometabolic risk factors were assessed including lipid profiles, glucose, blood pressure, and anthropometry. Linear mixed effects models were used for LINE-1, *H19*, and *11β-HSD-2* to account for the repeated-measure outcomes. Linear regression models were conducted for the cross-sectional association between *PPAR-α* with the outcomes. DNAm at LINE-1 was associated with log glucose at site 1 [β = −0.029, p = 0.0006] and with log high-density lipoprotein cholesterol at site 3 [β = 0.063, p = 0.0072]. *11β-HSD-2* DNAm at site 4 was associated with log glucose (β = −0.018, p = 0.0018). DNAm at LINE-1 and *11β-HSD-2* was associated with few cardiometabolic risk factors among youth in a locus-specific manner. These findings underscore the potential for epigenetic biomarkers to increase our understanding of cardiometabolic risk earlier in life.

**Keywords:** cardiometabolic risk factors; population-based study; children and adolescents; Mexicans; biomarkers; epigenetics; DNA methylation

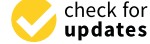



## 1. Introduction

Obesity is rising worldwide among children aged 5–19 years. In Latin America and the Caribbean region, prevalence rose over a 40-year period from 1.6% and 1.8% in 1975 to 10.4% and 13.4% in 2016 for girls and boys, respectively [1]. Obesity has been associated with increases in the risk and prevalence of cardiometabolic abnormalities among youth [2–4]. A cluster of cardiometabolic abnormalities, called metabolic syndrome [5,6], is considered a risk factor for cardiovascular disease (CVD) incidence, cardiovascular-related mortality, all-cause mortality [7,8], and other chronic diseases [9,10]. Rising prevalence of metabolic syndrome may be a driver of the CVD and type-2 diabetes epidemics [11]. Even though CVD outcomes are manifested in middle and late adulthood, cardiometabolic risk factors may become evident during childhood [12–17] and track into adulthood [4,18,19]. Identifying the determinants of cardiometabolic risk factors in youth could serve as a fundamental step for risk reduction and prevention [4,20].

Epigenetic modification is one potential underlying mechanism in obesity, cardiometabolic abnormalities, and CVD [21–28]. Previous research highlighted the importance of epigenetics as a potential biomarker for screening, diagnosis, prognosis, and individualized treatment regimens [23,24,29–31]. DNA methylation (DNAm), a commonly studied epigenetic modification, has been associated with CVD [21–27,32,33] and cardiometabolic risk factors, mainly in adults [21,28,34–36]. Existing evidence showed that DNAm during early development was associated with obesity and CVD risk later in life [37]; early embryogenesis is a particularly sensitive time period for epigenetic alteration by environmental factors that may contribute to disease risk [38]. However, adolescence is also a susceptible period for the impact of environmental stimuli on DNAm [39,40]. Furthermore, adolescence is characterized by changes in body composition and hormonal milieu [41]—the hallmarks for cardiometabolic abnormalities [42]. Despite the importance of this milestone, scare evidence is available investigating the potential of using DNAm as biomarkers for cardiometabolic health among children and adolescents.

The current study will address this gap in knowledge by examining the association of DNAm in blood leukocytes with cardiometabolic risk factors among Mexican children and adolescents enrolled in the Early Life Exposures in Mexico to ENvironmental Toxicants (ELEMENT) Cohort. Specifically, we quantified CpG site-specific DNAm at repetitive elements (long interspersed nuclear element-1, LINE-1), which comprises 15–17% of the human genome [43,44]. DNAm of LINE-1 is often used as a proxy measure for global DNAm [45], and it has been found to associate with CVD independent from well-established CVD risk factors in adults [46]. The other three genes are *H19*, 11β -hydroxysteroid dehydrogenase type 2 (*11β-HSD-2*), and peroxisome proliferator-activated receptor alpha (*PPAR-α*), which were selected based on their associations with components of cardiometabolic health. *H19* is an imprinted gene with a role in regulating cell formation and proliferation, body weight, adipogenesis, and brown adipose tissue thermogenesis [47–50], and abnormal fat partitioning is a crucial underlying factor in impaired cardiometabolic health [51]. *11β-HSD-2* converts cortisol to an inactive metabolite called cortisone [52,53]. Previous studies have associated *11β-HSD-2* regulation with blood pressure [54–56], insulin sensitivity [57], and type 2 diabetes [58]. Blood pressure and glucose hemostasis are cornerstones in assessing cardiometabolic health; however, the latter is of great interest for Hispanic youth as insulin resistance was reported among normal-weight Mexican youth [59]. Lastly, *PPAR-α* controls multiple lipid metabolism pathways [60,61], and it was associated with serum triglycerides [62]. *PPAR-α* dysregulation is thought to play a role in dyslipidemia, diabetes, and obesity [63]. Based on functions of the genes and results from related studies, we hypothesized that altered DNAm of these regions would associate with cardiometabolic health measures (waist circumference, blood pressure, and serum glucose, high-density lipoprotein cholesterol, and triglycerides) in children and adolescents.

## 2. Results

We assessed DNAm and cardiometabolic outcomes at one to two time points each in children from the ELEMENT cohort. The final sample sizes for LINE-1, *11β-HSD-2*, and *H19* were 242, 229, and 245 subjects, respectively, with DNAm at Time 1 and outcomes at Time 1 and/or Time 2. For *PPAR-α*, 345 subjects had DNAm and outcomes at Time 2 (Figure 1). Table 1 shows the demographic characteristics of the 410 children by time point. At Time 1, the mean (standard deviation (SD)) age of the sample was 10.34 (1.67) years and 53.25% were female. At Time 2, the mean age was 14.08 (2.03) years and 51.32% were female. Among cardiometabolic risk factors, only waist circumference and serum triglycerides values were greater at Time 2 than at Time 1. We examined the crude association between DNAm values across sites within each genomic region. We found that LINE-1, *H19*, and *PPAR-α* were moderately to strongly correlated with one another. *11β-HSD-2* sites were less correlated as we have observed in past studies with this same gene (Supplementary Tables S1–S4).

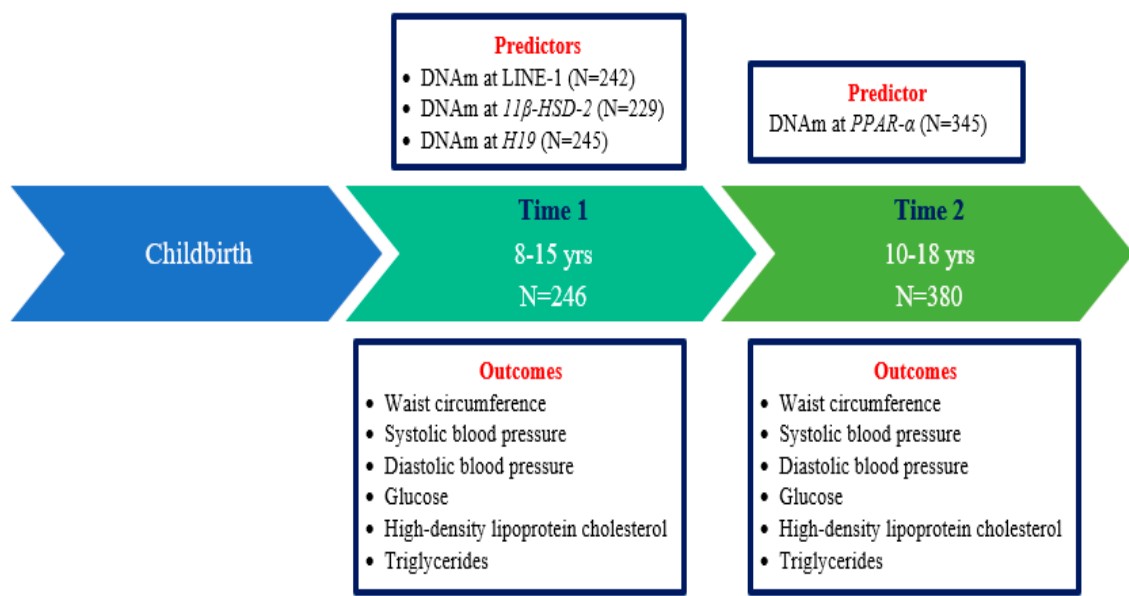

**Figure 1.** Summary of the Main Predictors and Outcomes for this Study and Number of Participants with the Data from the Early Life Exposures in Mexico to ENvironmental Toxicants (ELEMENT) Cohort. Abbreviations: DNAm = DNA methylation; Long interspersed nuclear elements (LINE-1); 11β-hydroxysteroid dehydrogenase type 2 (*11β-HSD-2*); Peroxisome proliferator-activated receptor alpha (*PPAR-α*).

**Table 1.** Descriptive Statistics of Mother and Child Characteristics of the Early Life Exposures in the Mexico to ENvironmental Toxicants (ELEMENT) Analytical Sample.

| | Time 1<br>*n* = 246 | Time 2<br>*n* = 380 |
|---|---|---|
| **Maternal Characteristics (At Time of Child's Birth)** | | |
| Years of education, % | | |
| <12 years | 121 (49.19) [1] | 196 (51.58) [2] |
| 12 years | 90 (36.59) [1] | 131 (34.47) [2] |
| >12 years | 34 (13.82) [1] | 52 (13.68) [2] |
| Age at childbirth, (years) | 26.86 (5.64) [1] | 26.47 (5.46) [2] |
| Parity, % | | |
| 1 | 90 (36.59) [1] | 144 (37.89) [2] |
| 2 | 89 (36.18) [1] | 135 (35.53) [2] |
| ≥3 | 66 (26.83) [1] | 100 (26.32) [2] |
| **Marital status, %** | | |
| Married | 175 (71.14) [1] | 274 (72.11) [2] |
| Other | 70 (28.46) [1] | 105 (27.63) [2] |
| **Enrollment in calcium supplementation study, %** | | |
| Not enrolled | 152 (61.79) [1] | 257 (67.63) [2] |
| Enrolled | 93 (37.80) [1] | 122 (32.11) [2] |
| **Child Characteristics (At birth)** | | |
| Female, % | 131 (53.25) | 195 (51.32) |
| Gestation age, (weeks) | 38.85 (1.49) [3] | 38.79 (1.61) [4] |
| Mode of delivery, % | | |
| Vaginal delivery | 140 (56.91) [5] | 220 (57.89) [6] |
| C-Section | 103 (41.87) [5] | 158 (41.58) [6] |
| Birth weight, (kg) | 3.15 (0.45) [7] | 3.15 (0.48) [6] |
| Breastfeeding duration, (months) | 8.15 (5.91) [1] | 8.09 (6.07) [2] |
| **Child Characteristics (At follow-up visits)** | | |
| Age, (years) | 10.34 (1.67) | 14.08 (2.03) |
| Body mass index Z-score for age | 0.85 (1.24) | 0.53 (1.26) [6] |
| Metabolic equivalents, (METs/week) | 31.38 (19.97) | 60.63 (38.76) |
| Total caloric intake, (kcal/day) | 2636.32 (839.83) | 2371.62 (936.82) |
| Pubertal onset, % | 103 (41.87) | 350 (92.11) [8] |

**Table 1.** *Cont.*

|  | Time 1 $n$ = 246 | Time 2 $n$ = 380 |
|---|---|---|
| Cardiometabolic risk factors (Outcomes) | | |
| Waist circumference, (cm) | 70.81 (10.71) | 79.14 (11.42) |
| Systolic blood pressure, (mmHg) | 102.74 (10.24) | 97.23 (9.62) |
| Diastolic blood pressure, (mmHg) | 65.58 (7.31) | 62.24 (6.71) |
| Fasting glucose, (mg/dL) | 86.98 (9.38) | 77.48 (7.05) [9] |
| High-density lipoprotein cholesterol, (mg/dL) | 58.76 (11.92) | 42.95 (8.87) [9] |
| Triglycerides, (mg/dL) | 87.89 (44.40) | 105.81 (57.47) [9] |
| DNAm (Predictors) | | |
| LINE-1 DNAm, % (averaged across 4 CpG sites) | 78.49 (2.31) [5] | N/A |
| *11β-HSD-2* DNAm, % (averaged across 5 CpG sites) [a] | −0.85 (1.34) | N/A |
| *H19* DNAm, % (averaged across 4 CpG sites) | 58.31 (5.16) [1] | N/A |
| *PPAR-α* DNAm, % (averaged across 2 CpG sites) | N/A | 10.62 (2.09) [10] |

Means (SD) or count (percentages) are presented for continuous or categorical variables, respectively. Number of missing values [1] $n$ = 245, [2] $n$ = 379, [3] $n$ = 242, [4] $n$ = 377, [5] $n$ = 243, [6] $n$ = 378, [7] $n$ = 244, [8] $n$ = 373, [9] $n$ = 342, [10] $n$ = 358. [a] Negative values appear for *11β-HSD-2* because values are standardized to controls included on each plate to reduce the impact of pyrosequencing batch effects. Abbreviations: DNAm = DNA methylation; Long interspersed nuclear elements (LINE-1); 11β-hydroxysteroid dehydrogenase type 2 (*11β-HSD-2*); Peroxisome proliferator-activated receptor alpha (*PPAR-α*).

### 2.1. Associations between the DNAm z-Score at LINE-1 and Repeated Measures of Cardiometabolic Risk Factors

In adjusted models, LINE-1 methylation levels were inversely associated with log serum fasting glucose at site 1 [β = −0.029, $p$ = 0.0006]; for each one standard deviation increase in DNAm (i.e., +4%), there was an approximately 3% decrease in log fasting glucose. In addition, a positive association was detected between DNAm at site 3 and log serum fasting high-density lipoprotein cholesterol [β = 0.063, $p$ = 0.0072], which means that for each one standard deviation increase in DNAm (i.e., +3%), there was an approximately 6% increase in log high-density lipoprotein cholesterol (Table 2). Sensitivity analysis (i.e., additionally adjusting for pubertal onset) did not attenuate the detected associations (Supplementary Table S5).

### 2.2. Associations between the DNAm z-Score at 11β-HSD-2 and Repeated Measures of Cardiometabolic Risk Factors

DNAm at site 4 showed an inverse association with log serum fasting glucose (mg/dL) [β = −0.018, $p$ = 0.0018] (Table 3). In sensitivity analysis (i.e., additionally adjusting for pubertal onset), associations found with fasting glucose maintained similar magnitude and significance (Supplementary Table S6).

### 2.3. Associations between the DNAm z-Score at H19 and Repeated Measures of Cardiometabolic Risk Factors

In the adjusted models, DNAm at none of the individual CpG sites was associated with the cardiometabolic outcomes (Supplementary Table S7). Results did not change in the two sensitivity analyses (Supplementary Tables S8 and S9).

### 2.4. Cross-Sectional Associations between the DNAm z-Score at PPAR-α and Cardiometabolic Risk Factors

In a cross-sectional analysis, DNAm was not associated with the cardiometabolic risk factors (Table 4). The sensitivity analysis (i.e., after adjusting for pubertal onset) showed the same result (Supplementary Table S10).

As an explanatory analysis, we assessed the crude association between DNAm values and gene expression for *PPAR-α*. RNA-seq data were available for a small subset of subjects at the same time point (i.e., Time 3) ($n$ = 65). Weak non-significant positive correlations were identified between mRNA and DNAm (site 1: Spearman's correlation [rs] = 0.14, ($p$ = 0.26); site 2: rs = 0.10, ($p$ = 0.42); average of the two sites rs = 0.12, ($p$ = 0.33)).

**Table 2.** Associations between the DNAm z-score at LINE-1 and Repeated Measures of Cardiometabolic Risk Factors using Mixed-effects Models (*n* = 242).

| | LINE-1 z-Score at Site 1 | | LINE-1 z-Score at Site 2 | | LINE-1 z-Score at Site 3 | | LINE-1 z-Score at Site 4 | |
|---|---|---|---|---|---|---|---|---|
| | **Estimate (SE)** | *p*-**Value** | **Estimate (SE)** | *p*-**Value** | **Estimate (SE)** | *p*-**Value** | **Estimate (SE)** | *p*-**Value** |
| Waist circumference (cm) (Total number of observations = 441, of which 43 (17.77%) subjects had one measurement and 199 (82.23%) subjects had two measurements) | | | | | | | | |
| Model 1 | −0.5960 (1.0435) | 0.5684 | 1.1418 (1.4217) | 0.4227 | −0.4783 (1.1510) | 0.6781 | 0.2997 (0.9013) | 0.7398 |
| Model 2 | 0.5615 (1.0072) | 0.5777 | 0.9837 (1.3686) | 0.4730 | −1.7757 (1.1106) | 0.1111 | 0.3214 (0.8710) | 0.7124 |
| Systolic blood pressure (mmHg) (Total number of observations = 441, of which 43 (17.77%) subjects had one measurement and 199 (82.23%) subjects had two measurements) | | | | | | | | |
| Model 1 | −0.4560 (0.8541) | 0.5939 | −0.1855 (1.1698) | 0.8741 | 0.1632 (0.9435) | 0.8628 | 0.9703 (0.7361) | 0.1887 |
| Model 2 | −0.9634 (0.8928) | 0.2817 | −0.00023 (1.2181) | 0.9999 | 0.4640 (0.9898) | 0.6397 | 0.8922 (0.7676) | 0.2464 |
| Diastolic blood pressure (mmHg) (Total number of observations = 441, of which 43 (17.77%) subjects had one measurement and 199 (82.23%) subjects had two measurements) | | | | | | | | |
| Model 1 | −0.5185 (0.5769) | 0.3697 | −0.1316 (0.7927) | 0.8682 | 0.2271 (0.6379) | 0.7221 | 0.3619 (0.4966) | 0.4669 |
| Model 2 | −0.6759 (0.5947) | 0.2570 | −0.04549 (0.8136) | 0.9555 | 0.3404 (0.6613) | 0.6072 | 0.3674 (0.5094) | 0.4716 |
| Log-transformed fasting glucose (mg/dL) (Total number of observations = 438, of which 46 (19.01%) subjects had one measurement and 196 (80.99%) subjects had two measurements) | | | | | | | | |
| Model 1 | −0.01570 (0.007838) | 0.0463 | 0.02427 (0.01086) | 0.0263 | −0.00357 (0.008708) | 0.6825 | −0.00361 (0.006726) | 0.5917 |
| Model 2 | −0.02864 (0.008211) | 0.0006 * | 0.02729 (0.01124) | 0.0160 | 0.01135 (0.009149) | 0.2162 | −0.00142 (0.007028) | 0.8402 |
| Log-transformed high-density lipoprotein cholesterol (mg/dL) (Total number of observations = 438, of which 46 (19.01%) subjects had one measurement and 196 (80.99%) subjects had two measurements) | | | | | | | | |
| Model 1 | 0.02078 (0.01893) | 0.2733 | −0.02664 (0.02610) | 0.3083 | 0.01023 (0.02099) | 0.6265 | −0.01677 (0.01627) | 0.3039 |
| Model 2 | −0.01466 (0.02111) | 0.4881 | −0.02801 (0.02873) | 0.3306 | 0.06331 (0.02334) | 0.0072 * | −0.00571 (0.01822) | 0.7543 |
| Log-transformed triglycerides (mg/dL) (Total number of observations = 438, of which 46 (19.01%) subjects had one measurement and 196 (80.99%) subjects had two measurements | | | | | | | | |
| Model 1 | −0.05170 (0.04055) | 0.2035 | −0.03424 (0.05541) | 0.5372 | 0.05445 (0.04481) | 0.2255 | −0.00392 (0.03498) | 0.9109 |
| Model 2 | −0.02698 (0.03945) | 0.4947 | −0.04343 (0.05383) | 0.4205 | 0.05072 (0.04378) | 0.2477 | 0.009633 (0.03392) | 0.7766 |

Long interspersed nuclear elements (LINE-1). Model 1 included LINE-1 z-scores at CpG sites 1, 2, 3, and 4 as fixed effects and a compound symmetry matrix structure to model the covariance structure of the repeated measurements for each outcome. Model 2 was additionally adjusted for the following fixed effects: age, sex, and duration of breastfeeding. * *p* < 0.008.

**Table 3.** Associations between the DNAm z-score at *11β-HSD-2* and Repeated Measures of Cardiometabolic Risk Factors using Mixed-effects Models (*n* = 229).

| | *11β-HSD-2* z-Score at Site 1 | | *11β-HSD-2* z-Score at Site 2 | | *11β-HSD-2* z-Score at Site 3 | | *11β-HSD-2* z-Score at Site 4 | | *11β-HSD-2* z-Score at Site 5 | |
|---|---|---|---|---|---|---|---|---|---|---|
| | Estimate (SE) | *p*-Value | Estimate (SE) | *p*-Value | Estimate (SE) | *p*-Value | Estimate (SE) | *p*-Value | Estimate (SE) | *p*-Value |
| Waist circumference (cm) (Total number of observations = 415, of which 43 (18.78%) subjects had one measurement and 186 (81.22%) subjects had two measurements) | | | | | | | | | | |
| Model 1 | −0.3822 (1.0424) | 0.7142 | −0.08657 (0.7980) | 0.9137 | 0.2635 (0.9701) | 0.7862 | 0.5264 (0.7690) | 0.4943 | 0.2132 (0.7252) | 0.7690 |
| Model 2 | −1.1319 (1.0012) | 0.2595 | 0.2204 (0.7707) | 0.7751 | 0.6173 (0.9303) | 0.5076 | 0.5578 (0.7361) | 0.4493 | −0.1382 (0.6969) | 0.8430 |
| Systolic blood pressure (mmHg) (Total number of observations = 415, of which 43 (18.78%) subjects had one measurement and 186 (81.22%) subjects had two measurements) | | | | | | | | | | |
| Model 1 | −1.6096 (0.8326) | 0.0545 | −0.7568 (0.6372) | 0.2362 | 1.2770 (0.7754) | 0.1010 | 0.3766 (0.6161) | 0.5416 | −0.4901 (0.5780) | 0.3974 |
| Model 2 | −1.4026 (0.8695) | 0.1083 | −0.7320 (0.6688) | 0.2751 | 1.1599 (0.8074) | 0.1524 | 0.3305 (0.6404) | 0.6064 | −0.3520 (0.6029) | 0.5600 |
| Diastolic blood pressure (mmHg) (Total number of observations = 415, of which 43 (18.78%) subjects had one measurement and 186 (81.22%) subjects had two measurements) | | | | | | | | | | |
| Model 1 | −0.9251 (0.5519) | 0.0951 | −0.8601 (0.4222) | 0.0428 | 0.3540 (0.5143) | 0.4920 | 0.4535 (0.4092) | 0.2690 | −0.01360 (0.3827) | 0.9717 |
| Model 2 | −0.8686 (0.5624) | 0.1240 | −0.8775 (0.4322) | 0.0436 | 0.3201 (0.5221) | 0.5404 | 0.4519 (0.4148) | 0.2771 | 0.01427 (0.3888) | 0.9708 |
| Log-transformed fasting glucose(mg/dL) (Total number of observations = 412, of which 46 (20.09%) subjects had one measurement and 183 (79.91%) subjects had two measurements) | | | | | | | | | | |
| Model 1 | −0.00076 (0.007513) | 0.9193 | 0.001955 (0.005764) | 0.7348 | 0.006329 (0.006998) | 0.3668 | −0.01869 (0.005586) | 0.0010 * | 0.002692 (0.005216) | 0.6064 |
| Model 2 | 0.009223 (0.007893) | 0.2440 | −0.00184 (0.006079) | 0.7624 | 0.001102 (0.007320) | 0.8805 | −0.01837 (0.005817) | 0.0018 * | 0.007427 (0.005472) | 0.1762 |
| Log-transformed high-density lipoprotein cholesterol (mg/dL) (Total number of observations = 412, of which 46 (20.09%) subjects had one measurement and 183 (79.91%) subjects had two measurements) | | | | | | | | | | |
| Model 1 | 0.002550 (0.01874) | 0.8919 | −0.00550 (0.01438) | 0.7026 | −0.00829 (0.01745) | 0.6351 | −0.01132 (0.01390) | 0.4161 | 0.005434 (0.01303) | 0.6770 |
| Model 2 | 0.02693 (0.02073) | 0.1952 | −0.02151 (0.01596) | 0.1793 | −0.02199 (0.01925) | 0.2545 | −0.01714 (0.01524) | 0.2620 | 0.01880 (0.01442) | 0.1938 |
| Log-transformed triglycerides (mg/dL) (Total number of observations = 412, of which 46 (20.09%) subjects had one measurement and 183 (79.91%) subjects had measurements) | | | | | | | | | | |
| Model 1 | 0.02425 (0.04126) | 0.5572 | 0.03580 (0.03163) | 0.2588 | 0.004623 (0.03838) | 0.9042 | 0.01794 (0.03047) | 0.5566 | −0.00972 (0.02872) | 0.7354 |
| Model 2 | 0.01469 (0.04003) | 0.7140 | 0.03065 (0.03084) | 0.3212 | 0.01000 (0.03715) | 0.7880 | 0.01977 (0.02946) | 0.5029 | −0.01685 (0.02782) | 0.5453 |

11β-hydroxysteroid dehydrogenase type 2 (*11β-HSD-2*). Model 1 included *11β-HSD-2* z-scores for CpG sites 1, 2, 3, 4, and 5 as fixed effects and a compound symmetry matrix structure to model the covariance structure of the repeated measurements for each outcome. Model 2 was additionally adjusted for the following fixed effects: age, and sex. * $p < 0.008$.

**Table 4.** Cross-sectional Associations between DNAm z-scores at *PPAR-α* and Cardiometabolic Risk Factors using Linear Regression (*n* = 345).

| | *PPAR-α* z-Score at Site 1 | | *PPAR-α* z-Score at Site 2 | |
|---|---|---|---|---|
| | Estimate (SE) | *p*-Value | Estimate (SE) | *p*-Value |
| Waist circumference (cm) (*n* = 345) | | | | |
| Model 1 | 0.71915 (0.71474) | 0.3150 | −1.70941 (0.65445) | 0.0094 |
| Model 2 | 0.99917 (0.70529) | 0.1575 | −1.68127 (0.64618) | 0.0097 |

**Table 4.** *Cont.*

| | *PPAR-α* z-Score at Site 1 | | *PPAR-α* z-Score at Site 2 | |
|---|---|---|---|---|
| | **Estimate (SE)** | *p*-**Value** | **Estimate (SE)** | *p*-**Value** |
| Systolic blood pressure (mmHg) (*n* = 345) | | | | |
| Model 1 | 0.58582 (0.60305) | 0.3320 | −1.02922 (0.55218) | 0.0632 |
| Model 2 | 0.49623 (0.57982) | 0.3927 | −0.66490 (0.53123) | 0.2116 |
| Diastolic blood pressure (mmHg) (*n* = 345) | | | | |
| Model 1 | 0.58530 (0.42242) | 0.1668 | −0.57466 (0.38679) | 0.1383 |
| Model 2 | 0.58072 (0.40724) | 0.1548 | −0.34026 (0.37311) | 0.3624 |
| Log-transformed fasting glucose (mg/dL) (*n* = 310) | | | | |
| Model 1 | 0.00598 (0.00614) | 0.3305 | 0.00016627 (0.00600) | 0.9779 |
| Model 2 | 0.00282 (0.00609) | 0.6443 | 0.00159 (0.00596) | 0.7900 |
| Log-transformed high-density lipoprotein cholesterol (mg/dL) (*n* = 310) | | | | |
| Model 1 | −0.00813 (0.01303) | 0.5329 | 0.01206 (0.01273) | 0.3445 |
| Model 2 | −0.00419 (0.01309) | 0.7490 | 0.00857 (0.01280) | 0.5035 |
| Log-transformed triglycerides (mg/dL) (*n* = 310) | | | | |
| Model 1 | 0.01232 (0.03058) | 0.6873 | 0.00118 (0.02989) | 0.9684 |
| Model 2 | 0.02086 (0.03057) | 0.4956 | −0.01116 (0.02989) | 0.7092 |

Peroxisome proliferator-activated receptor alpha (*PPAR-α*). Model 1 included PPAR-α z-scores for CpG sites 1 and 2. Model 2 was additionally adjusted for age, and sex.

## 3. Discussion

In this study, the relationships between DNAm at LINE-1, *11β-HSD-2, H19*, and *PPAR-α* with cardiometabolic risk factors were investigated among Mexican children and adolescents enrolled in a well-characterized birth cohort from Mexico City. Among cardiometabolic components, fasting glucose and high-density lipoprotein cholesterol were associated with DNAm of at least one genomic region. To the best of our knowledge, this is the first study investigating the potential of DNAm as a biomarker for cardiometabolic risk factors among Mexican youth using hypothesis-driven genomic regions.

The inverse and positive associations between LINE-1 DNAm and glucose and high-density lipoprotein cholesterol are in line with current evidence linking LINE-1 hypomethylation with genomic instability and CVD [46,64–66]. Furthermore, few studies conducted on adult populations showed inverse relationships between LINE-1 DNAm and impaired carbohydrate metabolism [67] and fasting glucose [62,68]. Scare and inconsistent evidence is available among pediatric populations with regard to cardiometabolic health and LINE-1 DNAm [69,70], where an inverse association detected with the waist circumference z-score [69] and null associations were reported with adiposity markers [70]. We acknowledge the complexity of crude comparisons across the studies because of the mismatch in the study endpoints and sample characteristics; therefore, future prospective studies are needed to strengthen the use of LINE-1 DNAm as a proxy for cardiometabolic health among youth.

We found that a one standard deviation increase in *11β-HSD-2* DNAm at site 4 (i.e., +2%) was associated with a decrease of 2% in fasting glucose. Our results could be explained

in light of the limited studies that investigated the connection between *11β-HSD-2* and glucose metabolism in adult populations [57,58]. Müssig et al. reported inverse association between 1β-HSD2 activity and insulin sensitivity [57], and Jang and colleagues found higher 11β-HSD2 enzyme activity among subjects with type 2 diabetes [58]. It is worth noting that not only is 11β-HSD-2 expression regulated by other epigenetic modifications [71], age [72], and lifestyle factors [73], but a lack of association was also documented earlier between 11β-HSD2 enzyme activity and mRNA expression [58]. As our results showed the potential of *11β-HSD-2* DNAm as a cardiometabolic biomarker among youth, future studies are needed combining DNAm, gene expression, and enzyme activity assessment to strengthen the evidence for the role of *11β-HSD-2* in cardiometabolic risk.

The present study has multiple strengths, including the prospective assessment of the association between DNAm at four genomic regions and up to two repeated measures of cardiometabolic risk factors during a sensitive period of growth, development, and maturation. We used a robust statistical model to account for the longitudinal data structure and conducted site-specific analyses for examining the association between the DNAm of each region with cardiometabolic risk factors. Site-specific approaches may be better when the data are not as correlated or when some CpG sites are much more variable than others in order to capture the complexity of the data. Our data come from a well-characterized birth cohort, ELEMENT, which allowed for assessing whether any of the mother's sociodemographic and reproductive characteristics would be potential confounding factors to account for. Furthermore, peripheral blood was used to quantify the DNAm because blood is an accessible tissue and commonly collected in clinical setting and epidemiological studies [31], which is a strength for investigating potential biomarkers for cardiometabolic risk factors among children.

With regard to the study limitations, the use of bisulfite treatment to measure DNAm does not distinguish between cytosine methylation (5mC) and cytosine hydroxymethylation (5hmC) [74], and 5hmC has its own distinct impact on gene regulation, which was not captured by our method. Therefore, the DNAm values might be confounded by hydroxymethylation because both 5hmC and 5mC are captured in the total DNAm percentage. Future studies should apply laboratory techniques that allow for distinguishing between 5hmC and 5mC. Additionally, our work has the limitation of including only DNAm without gene expression data for three of the four regions assessed. Because gene expression could be influenced by multiple factors, including other epigenetic modifications, physiological conditions, and lifestyle factors, we recommend future studies supplement the assessment of DNAm with gene expression and carefully take into account the other potential factors that influence gene expression. Such evidence will strengthen the use of DNAm as a clinical biomarker for cardiometabolic health if clinical validation studies confirm its utility.

We acknowledge the age heterogeneity as our analysis includes pre-teenagers and teenagers; given our small sample size, we did not explore the relationships stratified by age groups. Thus, future studies are needed to investigate the potential role of age in modifying the association between DNAm and cardiometabolic risk factors during pubertal transition. Additionally, the magnitude of detected associations was small, which might not be of clinical significance. However, small effect sizes are typically reported in epigenetic studies [62,65,67,68,75]. Because small effects may still have relevance for children's health outcomes [75], further studies are needed to enhance our understanding of the cause-and-effect relationship between DNAm and cardiometabolic health by validating our results in independent large-scale population-based youth populations with objective assessment of lifestyle patterns known to influence DNAm. Such evidence will facilitate the progress toward increasing the reproducibility and strengthening the biological relevance of DNAm biomarkers. Additionally, despite our consideration for addressing multiple testing, we still acknowledge the possibility of reporting false positive results due to chance. Lastly, the possibility of residual confounding—such as smoking status and genetic variants—and reverse causation between DNAm and cardiometabolic outcomes cannot be ruled out.

## 4. Materials and Methods

### 4.1. Study Population

The analytical sample consisted of offspring who participated in two of three sequentially enrolled birth cohorts of the ELEMENT project in Mexico City, Mexico. A comprehensive description of the ELEMENT project and the eligibility and exclusion criteria are available elsewhere [76]. Briefly, the ELEMENT project included mother–child dyads recruited from maternity hospitals representing women from low- to middle-income population groups from 1997 to 2005 [77]. Mothers recruited for one of the birth cohorts were enrolled in a randomized controlled trial (RCT) that examined the role of daily calcium supplementation during pregnancy (1200 mg/day) in mitigating the effect of lead exposure on the neurobehavioral and physical developmental outcomes in offspring [76]. Offspring were followed at multiple time points in childhood and through adolescence; the aim of the follow-up visits was to follow as many children from the original birth cohort as possible, prioritizing younger ages at specific time points. The sample size for each follow-up visit was determined by the aims for the original grant-funded visit.

We utilized available data from two follow-up visits. In the first follow-up visit, herein called Time 1, we planned to follow 250 children aged between 8 and 15 years. We prioritized children according to availability of prenatal biological samples for offspring from the original birth cohorts [76]. The second follow up visit, Time 2, was conducted on average 2 years later (maximum time to follow-up was 4.6 years). We planned to follow >500 children from the original birth cohorts. We prioritized the 250 subjects from the Time 1 visit (of which a large majority (~90%) returned) and added additional ELEMENT children who were not included in the Time 1 visit. Based on a statistical power calculation and available funds, we selected a sub-sample of these for epigenetic analysis (all children at Time 1 and >350 at Time 2). Children were 10–18 years of age at Time 2.

The analytical sample for the genomic regions LINE-1, *H19*, and *11β-HSD-2* included children and adolescents who had DNAm data at Time 1 and had data for at least one of the six cardiometabolic risk factors (i.e., waist circumference, systolic and diastolic blood pressure, fasting glucose, triglycerides, high density lipoprotein cholesterol) at Time 1 and/or Time 2. DNAm at *PPAR-α* was measured only at Time 2; subjects with these data and at least one of the six cardiometabolic risk factors were included for the analytical sample for *PPAR-α* models. The National Institute of Public Health of Mexico and the University of Michigan institutional review boards approved the research protocols. Written informed consents were collected from mothers upon their enrollments and assent from adolescents.

### 4.2. Laboratory Measurements and Outcomes

#### 4.2.1. DNA Methylation Analysis

The current study limits its focus to four genomic regions, which have previously been associated with cardiometabolic risk factors. Whole blood samples were collected via venipuncture into tubes containing ethylenediaminetetraacetic acid (EDTA) preservative (Paxgene and BD Vacutainer) by trained staff following standard protocols. High-molecular-weight DNA was extracted from blood leukocytes with the PAXgene Blood DNA kit (PreAnalytix, Switzerland) or the Flexigene kit (Qiagen). The extracted DNA samples were treated with sodium bisulfite using Epitect (Qiagen, Valencia, CA, USA) or EZ DNA Methylation kits (Zymo Research, Irvine, CA, USA) following the standard methods previously published [78]. The purpose of bisulfite treatment was to convert the un-methylated cytosines to uracil and to preserve the methylated cytosines. The bisulfite-treated DNA samples were amplified using HotStarTaq Master Mix (Qiagen), and primers designed to amplify each region of interest. Pyrosequencing was performed using either PyroMark Q96 MD (Qiagen) or PyroMark Q96 ID (Qiagen). Pyro Q-CpG Software calculated the percent methylation and performed internal quality control checks. At Time 1, DNAm was quantified for *H19* (4 CpG sites in the imprinting control region), for LINE-1 (4 CpG sites in a conserved region across many LINE-1s), and for *11β-HSD-2* (5 CpG sites in the

promoter region) and at Time 2 for *PPAR-α* (2 CpG sites in the promoter region) following the protocols published previously [79–81]. Information on these genomic regions and the primer sequences is presented in Supplementary Table S11 [77]. More than 10% of all samples and controls of human DNA with known percentages of DNAm (0%, 25%, 50%, 75%, and 100%) were run in duplicate and included in each pyrosequencing batch (96-well plate). The average of duplicate samples was used when applicable [82]. DNAm data from LINE-1, *11β-HSD-2*, and *H19* suggested a batch effect, and the methylation percentages were standardized to adjust for the batch effects as described previously [82]. We then standardized DNAm values for each region to have mean 0 standard deviation 1 based on the sample's mean and standard deviation values to express the DNAm as a z-score, and these z-scores were used in statistical analysis.

Samples collected at Time 1 were not preserved for downstream RNA isolation. At Time 2, blood leukocytes preserved for RNA isolation were collected from all participants and archived. Of these, 72 were selected for next-generation sequencing of RNA ('RNA-Seq'). Samples were prioritized for selection that had the highest quality and quantity of RNA and had complete datasets needed for previous questions of interest [83]. Of those, 65 were from participants included in this manuscript. The read count of *PPAR-α* from the RNA-seq was used to assess the relationship between DNAm and gene expression for *PPAR-α*. The RNA-seq protocol followed was previously described [83].

### 4.2.2. Cardiometabolic Risk Factors

#### Anthropometric Measures

Duplicate measurements were collected by trained research staff for body weight to the nearest 0.1 kg using a digital scale (BAME Model 420; Catálogo Médico) and In-Body 230 (Biospace Co, Ltd, Seoul, Republic of Korea), height to the nearest 0.5 cm, and waist circumference to the nearest 0.1 cm using a non-stretchable measuring tape SECA (model 201, Hamburg, Germany) [84]. The average of the two measurements was used for the analysis [85]. These measurements were conducted at Time 1 and Time 2.

#### Blood Pressure Measurements

Duplicate readings of systolic and diastolic blood pressure were recorded in a seated position using a mercury sphygmomanometer (TXJ-10 MD 3000 model, Homecare, Nanjing, China), and the average of the two measurements was used for the analysis. These measurements were conducted at Time 1 and Time 2.

#### Fasting Biomarkers

At each follow-up visit (T1 and T2), trained research staff collected blood samples from children after an 8 h overnight fast. Fasting glucose and lipids were measured in serum at the Michigan Diabetes Research Center Chemistry Laboratory. Specifically, fasting glucose was assessed via automated chemiluminescence immunoassay (Immulite 1000; Siemens Medical Solutions). Triglycerides were quantified via an enzymatic colorimetric method using a Cobas Mira automated chemistry analyzer (Roche Diagnostics, Indianapolis, IN, USA). The level of high-density lipoprotein cholesterol was obtained by using direct high-density lipoprotein cholesterol (Roche Diagnostics) [85]. All serum markers were above the limit of detection (LOD).

### 4.3. Covariates

Based on prior knowledge of cardiovascular and metabolic health, covariates assessed for this research were classified as (1) maternal and child characteristics around the time of birth (sex, birth weight, gestation age, mode of delivery, duration of breastfeeding, and mothers' age, marital status, parity, years of education, and enrollment in the calcium supplementation study during pregnancy) and (2) follow-up characteristics for the children, which were measured at the baseline visit for each exposure, e.g., child's age, total caloric intake, physical activity measured as metabolic equivalents, and pubertal onset. In our

statistical analysis section, we explained our rationale for selecting covariates in each adjusted model.

After childbirth, mothers reported household and demographic information, including their ages, marital status (married compared to any other status), parity status (1, 2, $\geq$3), and years of education (<12 yrs, 12 yrs, or >12 yrs), gestational age estimated by a registered nurse, and mode of delivery (vaginal, or C-section childbirth). The newborns were followed until 5 years of age, and information about self-reported breastfeeding duration was estimated [86]. Since cohort 3 was an RCT for daily calcium supplementation during the first trimester of pregnancy until 1-year postpartum and cohort 2 participants were not part of a trial, we created a binary indicator for mothers who received the calcium treatment (yes/no) with all mothers from cohort 2 falling into the 'no' category [76,87].

During each of two follow-up visits, total caloric intake was quantified using a semi-quantitative food frequency questionnaire (FFQ) that captured the intake over the previous week [84,88]. The FFQ was adapted from the Mexican National Health and Nutrition Survey, and FFQs were analyzed using food composition software developed by the National Institute of Public Health, Mexico [89]. A physical activity questionnaire was developed based on the Youth Activity Questionnaire (YAQ) and validated relative to 24 h physical activity recall among Mexican school-children aged 10 to 14 years in Mexico City [90]. For each self-reported physical activity, the corresponding metabolic equivalent was multiplied by the activity intensity [91]. The total metabolic equivalents per week were calculated by summing the metabolic equivalents for all activities. Puberty was assessed through Tanner staging for breast and pubic hair (for girls) or genitalia and pubic hair (for boys) [92,93] by trained physicians [94]. Consistent with previous ELEMENT publications in which pubertal onset was a covariate, we classified children as having pubertal onset when the Tanner Stage for either or both of pubic hair and genital development (boys) or pubic hair and breast development (girls) was greater than one [95–97].

### 4.4. Statistical Analysis

Outcomes were cardiometabolic risk factors: waist circumference, systolic blood pressure, diastolic blood pressure, glucose, high-density lipoprotein cholesterol, and triglycerides. Dependent variables of interest were DNAm z-scores at LINE-1, *11β-HSD-2*, *H19*, and *PPAR-α* after standardizing the values based on the sample's mean and standard deviation for each site. Outcomes and exposures were treated as continuous in our models. The demographic characteristics of the study participants were presented as the mean (SD) and counts (proportions) for continuous and categorical variables, respectively.

DNAm percentages were quantified at multiple loci (CpG sites) within the same genomic region (i.e., *H19*: 4 CpG sites, LINE-1: 4 CpG sites, *11β-HSD-2*: 5 CpG sites, and *PPAR-α*:2 CpG sites). For each genomic region, the DNAm percentages at all CpG sites were included as repeated measures of the same variable in models of each outcome. To illustrate, the LINE-1 z-scores at CpG site 1, 2, 3, and 4 were included as four fixed effects in our models, and the same strategy was applied for other genes. This analytical approach was used in previous publications [98].

To examine the relationship between DNAm at Time 1 for LINE-1, *11β-HSD-2*, and *H19* and each cardiovascular risk factor outcome, separate linear mixed-effects models with a compound symmetry covariance structure were used to model the covariance structure of the repeated outcome assessed at Time 1 and 2. We used linear regression to assess the cross-sectional association between DNAm at *PPAR-α* and the outcomes because this gene was only measured at Time 2. For each exposure, the crude model included only DNAm z-scores at multiple CpG sites for a genomic region. Due to the biological plausibility for the sex and age difference in DNAm, we considered age and sex as mandatory covariates in any fully adjusted model. For the other covariates, we followed a parsimonious approach. Therefore, covariates were adjusted for only if they were potential confounders among our study population based on the significance of their statistical association with each gene of interest (i.e., $p < 0.05$). We investigated the confounding factors for each genomic region

by examining the distribution of childbirth and follow-up characteristics across quartiles of average DNAm z-scores of all loci within the region using either analysis of variance or Kruskal-Wallis H tests for continuous covariates that were normally and non-normally distributed, respectively, and a chi-squared test for categorical covariates. Based on these investigations to select the confounding factors, only LINE-1 DNAm was associated with breastfeeding duration. Therefore, LINE-1 models included breastfeeding duration, in addition to age and sex (Supplementary Tables S12–S15).

Our mixed-effects models' tables show information about the total sample size (i.e., number of unique subjects), total number of observations used in each model, and number of subjects with repeated measures for each outcome. Our linear regression models' tables show information about the total sample size for each outcome. Collinearity was assessed in the linear regression models using variance inflation factors. We conducted sensitivity analyses. First, we adjusted for the pubertal onset at Time 1 for LINE-1, *11β-HSD-2*, and *H19* and at Time 2 for *PPAR-α* because puberty has been associated with DNAm [39]. We also repeated the analysis after excluding one outlier value in DNAm for *H19*. The SAS statistical software package, version 9.4, was used for analyses (SAS Corp, Cary, NC), and a $p < 0.008$ was considered a statistically significant association following correction for multiple testing of six outcomes ($p < 0.008$ or 0.05/6).

## 5. Conclusions

In conclusion, we observed associations between DNAm at specific CpG sites for LINE-1 and glucose and high-density lipoprotein cholesterol and for *11β-HSD-2* and glucose in a sample of Mexican youth. Our finding supplemented existing knowledge on the potential of epigenetics to identify the molecular mechanism underlying cardiometabolic abnormalities, and it could open the door for targeted interventions among youth. Nevertheless, our results merit further investigation to replicate, validate, and expand on the use of DNAm though carefully designed prospective studies in multiple independent pediatric populations. Moreover, since our study only focused on four genomic regions, we recommend future studies employ epigenome-wide approaches to identify all important genes for these outcomes in youth.

**Supplementary Materials:** The following supporting information can be downloaded at: https://www.mdpi.com/article/10.3390/epigenomes7010004/s1, Table S1: Spearman's rank correlation coefficients between the DNAm z-scores at LINE-1 CpG sites; Table S2: Spearman's rank correlation coefficients between the DNAm z-scores at *11β-HSD-2* CpG sites; Table S3: Spearman's rank correlation coefficients between the DNAm z-scores at *H19* CpG sites; Table S4: Spearman's rank correlation coefficients between the DNAm z-scores at *PPAR-α* CpG sites; Table S5: Associations between the DNAm z-score at LINE- 1 and Repeated Measures of Cardiometabolic Risk Factors using Mixed-effects Models Adjusted for Pubertal Onset ($n = 242$); Table S6: Associations between the DNAm z-score at *11β-HSD-2* and Repeated Measures of Cardiometabolic Risk Factors using Mixed-effects Models Adjusted for Pubertal Onset ($n = 229$); Table S7: Associations between the DNAm z-score at *H19* and Repeated Measures of Cardiometabolic Risk Factors using Mixed-effects Models ($n = 245$); Table S8: Associations between the DNAm z-score at *H19* and Repeated Measures of Cardiometabolic Risk Factors using Mixed-effects Models after the Removal of Outlier DNAm Values ($n = 244$); Table S9: Associations between the DNAm z-score at *H19* and Repeated Measures of Cardiometabolic Risk Factors using Mixed-effects Models Adjusted for Pubertal Onset ($n = 245$); Table S10: Cross-sectional Associations between the DNAm z-score at *PPAR-α* and Cardiometabolic Risk Factors using Linear Regression Adjusted for Pubertal Onset ($n = 345$); Table S11: Primer Sequences and Details of CpG Sites Assessed; Table S12: Average DNAm z-score at LINE-1 and Confounder Selection; Table S13: Average DNAm z-score at *11β-HSD-2* and Confounder Selection; Table S14: Average DNAm z-score at *H19* and Confounder Selection; Table S15: Average DNAm z-score at *PPAR-α* and Confounder Selection.

**Author Contributions:** Data provision: J.M.G., D.C.D., K.E.P., A.C., M.M.T.-R. and L.A.T.-O.; conceptualization: A.A.A., J.M.G.,. K.E.P., H.M.K., E.A.R.-N. and A.B.; data analysis: A.A.A.; writing—original draft: A.A.A.; writing—review and editing: J.M.G. and K.E.P.; supervision: J.M.G. and K.E.P.

All authors provided critically important intellectual contribution to the content and have read. All authors have read and agreed to the published version of the manuscript.

**Funding:** Funding for the Early Life Exposure in Mexico to ENvironmental Toxicants (ELEMENT) was provided by the U.S. Environmental Protection Agency (US EPA) (RD83480019, RD83543601) and the National Institute for Environmental Health Sciences (NIEHS) (P20 ES018171, P01 ES02284401, and R35 ES031686). No additional financial support was received.

**Institutional Review Board Statement:** This study was conducted according to the guidelines of the Declaration of Helsinki and approved by the Institutional Review Boards of the University of Michigan and the National Institute of Public Health of Mexico.

**Informed Consent Statement:** Written informed consents were collected from mothers upon their enrollments in the ELEMENT project with assent from adolescents. No additional consent was required for publication.

**Data Availability Statement:** The datasets generated during and/or analyzed during the current study are not publicly available due to human subjects' rights, but the data are available upon reasonable request to the ELEMENT PI, Karen Peterson (karenep@umich.edu) for review by the ELEMENT committee.

**Acknowledgments:** We gratefully acknowledge the mothers and children who participated in the Early Life Exposure in Mexico to Environmental Toxicants (ELEMENT) study and American British Cowdray Medical Center (ABC) for providing facilities for this research.

**Conflicts of Interest:** The authors declare that there is no conflict of interest.

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
