# Peer review of "DNA Methylation Is a Potential Biomarker for Cardiometabolic Health in Mexican Children and Adolescents"

_2075-4655, 2022_

Round 1

Reviewer 1 Report

This manuscript, “DNA Methylation is a Potential Biomarker for Cardiometabolic Health in Mexican Children and Adolescents”, aimed to associate the methylation changes at certain genes/genomic regions (LINE-1, H19, 11β-HSD-2, and PPAR-α) with cardiometabolic risk factors in adolescents. Overall, the data analysis was appropriate, and the findings provided helpful insights.

The following comments or suggestions, if can be addressed, would further strengthen this manuscript.

  1. Could the authors discuss a little bit more about why pyrosequencing instead of array-based methylation measurements was used in the study? Array-based methylation (i.e., Illumina EPIC array) may provide a more comprehensive profile of the methylome.
  2. The authors measured several CpGs for each gene/genomic region. Based on the supplementary table 1, CpGs within each gene are close together (often < 20bp). Could the authors provide the correlation of these CpGs? One would assume the methylation levels of CpGs close together tend to be positively correlated.
  3. The authors mentioned that “The inverse and positive associations between LINE-1 DNAm and glucose and high-density lipoprotein cholesterol, respectively, are in line with current evidence linking LINE-1 hypomethylation with genomic instability and CVD”. However, this observation was made based on only 1 out of 4 CpGs measured in L1, and the effect size is relatively small. In addition, the other CpGs showed a lack of effect or even an opposite trend. It is possible that these significant associations are based on random chance only. It may be better to discuss this limitation in the discussion section.
  4. CpGs in 11β-HSD-2 and PPAR-α are located in the promoter region. Is there any evidence that the methylation levels of CpGs measured in the study associated with the expression of the corresponding genes?

Reviewer 2 Report

Aljahdali et al.'s paper detail an interesting investigation that revealed associations between DNAm and cardiometabolic risk factors in children and adolescents. While I think the topic is timely and exciting, I have a few suggestions and concerns regarding the way the paper is currently written:

  1) Abstract: you should include information about the cross-sectional study and gene-expression analysis.   2) Introduction: generally quite well written.   3) Methods:  3.1) Can you please provide further details of the follow-up Time 1? For Time 2, you mentioned that it was approximately three years later. However, you should have precise Time 1.  3.2) You mention that at least one of the six cardiometabolic risk factors was considered for the analysis. Does one factor sufficient for cardiovascular risk measurement?  3.3) Does DNAm not change during the longitudinal follow-up? Why did you measure DNAm PPAR-α only Time 2? The same for fasting lipids - why did it not include cholesterol?  3.4) Can you please provide further details of fasting biomarkers? Did you collect these biomarkers only once? It’s good to be more precise.   4) Results: 4.1) The authors used Z-score in specific variables such as DNAm at four genomic regions and BMI. However, they did not include this statistic for other variables. Why the authors opted for this statistical analysis? The same analysis was shown with measures of cardiovascular risk factors (log-transformed in fasting glucose, HDL and triglycerides, and crude values in waist circumference, systolic and diastolic blood pressure).  4.2) Why did you covary for sex and age in some associations between DNAm and outcomes, and in others, you included the duration of breastfeeding? Are there different influences for these outcomes considering your target genes?  4.3) You did a correlation between DNAm and gene expression for PPAR-α, but you found a weak significance. Is there a clinical significance?   5) Discussion: I suggest you explain if there are transcription factors in the selected four genomic regions and their clinical implications for your study.    6) Conclusion: Please be more specific. What are findings between DNAm and cardiovascular risk factors mean?
